# Genome-Wide CRISPR Screen Identifies Genes Involved in Metastasis of Pancreatic Ductal Adenocarcinoma

**DOI:** 10.3390/cancers16213684

**Published:** 2024-10-31

**Authors:** Risky Oktriani, Anna Chiara Pirona, Lili Kalmár, Ariani S. Rahadian, Beiping Miao, Andrea S. Bauer, Jörg D. Hoheisel, Michael Boettcher, Haoqi Du

**Affiliations:** 1Functional Genome Analysis, German Cancer Research Center (DKFZ), Im Neuenheimer Feld 580, 69120 Heidelberg, Germany; risky.oktriani@ugm.ac.id (R.O.); anna-chiara.pirona@weizmann.ac.il (A.C.P.); lili.kalmar@web.de (L.K.); ariani.rahadian@med.uni-heidelberg.de (A.S.R.); andrea.bauer@dkfz.de (A.S.B.); haoqidu@nwu.edu.cn (H.D.); 2Faculty of Biosciences, Heidelberg University, Im Neuenheimer Feld 234, 69120 Heidelberg, Germany; 3Department of Biochemistry, Faculty of Medicine, Public Health and Nursing, Universitas Gadjah Mada, Farmako Sekip Utara, Yogyakarta 55281, Indonesia; 4Mannheim University of Applied Sciences, Paul-Wittsack-Straße 10, 68163 Mannheim, Germany; 5Immune Regulation in Cancer, German Cancer Research Center (DKFZ), Im Neuenheimer Feld 580, 69120 Heidelberg, Germany; b.miao@dkfz-heidelberg.de; 6Medical Faculty, Martin Luther University Halle-Wittenberg, Kurt-Mothes-Straße 3a, 06120 Halle, Germany; michael.boettcher@medizin.uni-halle.de; 7School of Medicine, Faculty of Life Sciences and Medicine, Northwest University, 229 Taibai North Road, Xi’an 710069, China

**Keywords:** CRISPR-Cas9 knockout screen, metastasis, pancreatic cancer, transcription factors, genetic interaction

## Abstract

A CRISPR knockout screen was performed targeting all genes of the human genome with about 12 knockout sgRNAs each. By comparing pancreatic cancer cells with high or low metastatic capacity, genes were identified that affect the viability of metastatic cells while not affecting cell viability of non-metastatic cells. Further functional studies looked at some of the identified candidate genes. For one of the metastasis-related genes—*MYBL2*—a change in its interaction with another regulator gene was found to be implicated in metastasis.

## 1. Introduction

Pancreatic ductal adenocarcinoma (PDAC) is one of the deadliest tumour forms; its mortality is close to incidence [1,2,3]. This dismal situation is due to a lack of means to diagnose the disease early. It frequently develops without obvious symptoms, and tumours grow very aggressively and metastasize early. In addition, there is a lack of therapeutic modalities with major impact [4]; surgery is still the best option. Because of the usually advanced tumour stage at the time of diagnosis, however, less than 20% of patients can undergo tumour resection, and even they usually experience tumour recurrence. In the majority of PDAC patients, cancer cells have spread from the site of the primary tumour to distant organs at the time of diagnosis. While common driver mutations in genes *KRAS*, *CDKN2A*, *TP53*, and *SMAD4* have been observed [1], the overall picture in PDAC at the mutational, and even more so at the transcriptional level, is rather complex and varies substantially between patients. Even more complexity is introduced during the processes required for metastatic cancer cell dissemination and clonal expansion. While information about genetic variations exists [5,6], important molecular aspects of PDAC metastasis are not fully understood. Towards dealing better with pancreatic cancer metastasis, it is a prerequisite to unravel underlying molecular processes [7].

Towards these ends, knockout screens with clustered regularly interspaced short palindromic repeats (CRISPR)-associated protein 9 (Cas9) have proven to be a robust approach for identifying genes that may be crucial for cancer (e.g., [8,9]), meaning that the loss of their function reduces tumour cell viability substantially [10]. Here, we applied a highly complex, comprehensive CRISPR library to knock out the entire human gene set in two PDAC cell lines that are basically identical apart from their metastatic capacity. The objective was to identify genes that are essential for the survival of cells with metastatic features, while not being vital to non-metastasizing cells. Any such gene could provide an opportunity to specifically target metastatic tumour cells and, thus, affect this important disease mechanism. A small subset of identified candidate genes was analysed further concerning functional aspects, revealing different processes affecting metastasis. The gene encoding transcription factor MYBL2 was studied in more detail. No new gene functions were found to be involved, however. Interestingly, a change in the genetic interaction between *MYBL2* and the transcription factor gene *FOXM1* was identified that may be relevant for regulating the metastatic mode of pancreatic tumour cells.

## 2. Materials and Methods

### 2.1. Cells

S2-007 (RRID: CVCL_B279) and S2-028 (RRID: CVCL_B282) cells were kindly provided by Malte Buchholz (Marburg University, Marburg, Germany); culturing was in DMEM (Thermo Fisher Scientific, Waltham, MA, USA) supplemented with 10% FBS and penicillin/streptomycin (Thermo Fisher Scientific). HEK293T/17 (RRID: CVCL_1926) cells were obtained from ATCC and solely used for lentiviral packaging of the sgRNA library; culturing was in IMDM (Thermo Fisher Scientific) supplemented with 10% FBS and penicillin/streptomycin. All cells were authenticated and regularly tested for mycoplasma contamination.

### 2.2. CRISPR-Cas9 Screening

CRISPR screening was performed using a library of 259,900 single guide RNA (sgRNA) constructs, with about 12 sgRNAs per gene, as previously reported [11] (see also Appendix A and [12,13,14]). A multiplicity of infection of 0.3 and 100-fold coverage were used for transduction. Cells were collected at days 6 (t_zero_) and 24 (t_end_). DNA was isolated and used for next-generation sequencing. Total read counts for each sgRNA sequence were analysed with the MAGeCK-VISPR 0.5.6 software [15,16]. Kyoto Encyclopedia of Genes and Genomes (KEGG) pathways were assigned using DAVID version 6.8 ([17]; https://david.ncifcrf.gov/). Pathview in the R software package version 1.4.1106 was used to plot genes on the KEGG pathways. Network gene analysis was performed with the Ingenuity Pathway Analysis V6 software tool (Qiagen, Hilden, Germany).

### 2.3. Generation of Cells with Individual Gene Knockouts

Five sgRNAs of each candidate gene, two non-targeting control (NTC) constructs, and empty vector were separately transduced into cells; for sequences, see Appendix A. Selection with puromycin of positively transduced cells was performed after two days. The actual knockouts were confirmed by Western blot immunoassays.

### 2.4. Cell Viability Assay Using Flow Cytometry

Transduced cells were mixed with wild-type cells at a ratio of 4:1 for single transduction and 1:1 for double transduction. Cells were incubated in 6-well microtiter plates and harvested every 7 days. About a million cells were pelleted by centrifugation and washed with PBS. Ten thousand cells were analysed on a Guava EasyCyte HT flow cytometer (Luminex, Austin, TX, USA), taking advantage of the mCherry label in transduced cells.

### 2.5. Colony Formation Assay

Two millilitres of 0.5% SeaKem GTG Agarose (Lonza, Basel, Switzerland) in PBS were transferred to each well of 6-well microtiter plates. Top agar (0.05% agar) was prepared at 37 °C, mixing 3.2 mL cell culture medium, 0.4 mL FBS, and 0.4 mL 2% agar. One millilitre was mixed with 2500 cells and immediately transferred to a well filled with base agar. After agar solidification, 1 mL DMEM medium was added to each well. Incubation was for 15 days until colonies appeared. The medium was discarded and 0.5 mL of 0.005% crystal violet (Sigma Aldrich, Heidelberg, Germany) was added, followed by a 1 h incubation at room temperature. After washing with PBS, the colony number was counted using the Open CFU 3.8 software [18].

### 2.6. Migration Wound Healing Assay

The assay was performed in µ-dishes (ibidi, Gräfelfing, Germany), following the manufacturer’s protocol. Images of the cell gaps were acquired at the beginning of the incubation and after 3 days with an Axio Observer Z1 inverted microscope (Carl Zeiss, Oberkochen, Germany). The gap size was calculated using the ImageJ v1.5.3 software downloaded from the US NIH (https://imagej.net).

### 2.7. Invasion Assay

Cell invasion capacity was studied in microtiter plates with BioCoat Control Inserts with an 8.0 μm membrane (Corning, Corning, NY, USA) following the manufacturer’s protocol. Colony counting was performed using Open CFU 3.8 software [18].

### 2.8. Double Transduction

Cells were transduced with lentivirus containing sgRNAs targeting *FOXM1* or *MYBL2*; for sequences, see Appendix A. For this purpose, the plasmids pAW12.lentiguide.GFP and pAW13.lentiguide.mCherry [19] were used, which encode for different fluorescent proteins. Two days after transduction, cells were sorted using a FACSAria III sorter (BD Biosciences, Franklin Lakes, NJ, USA) to select double-positive transductions, taking advantage of the different wavelengths of GFP and mCherry. Data analysis was with FlowJo v10.8.1 software (BD Biosciences). Selected cells were cultured and characterized by Western blot immunoassays.

### 2.9. Cell Cycle Assay

Cells were plated in a 6-well plate. After a 3-day incubation, they were trypsinised, collected in a 15 mL falcon tube, and centrifuged at 1000 rpm for 5 min. After washing with PBS, cold 70% ethanol was used to fix the cells for at least 1 h. Cells were washed twice with PBS and resuspended in 150 µL PBS, the same volume of propidium iodide (Sigma Aldrich, Heidelberg, Germany), and 5 µL of 1 μg/µL RNAse A (Thermo Fisher Scientific). Cells were analysed using a BD FACSCanto flow cytometer (BD Biosciences).

### 2.10. In Vivo Colonisation Assay

Female NOD SCID gamma (NSG) mice (8–9 weeks old, 6 mice/group) were recruited from the DKFZ Centre for Preclinical Research and kept at a 12 h light–dark cycle with unrestricted Kliba 3307 diet and water. S2-007 cells with either *MYBL2* knockout or an NTC control were used. After randomisation, half a million cells in Ca^2+^-free PBS were injected into the lateral tail vein (i.v.) or, alternatively, 0.05 million cells into the left heart ventricle (i.card.), the latter under isoflurane anaesthesia. There was no blinding. Besides daily health checks, body weight was measured at least once weekly throughout the experiment. Time of necropsy was as indicated or when mice reached a pre-defined humane endpoint (weight loss ≥ 20%, breathing problems, increased abdominal volume, poor health status observed by macroscopic inspection). After cervical dislocation, dissected lung and liver sections were histologically analysed with polyclonal guinea pig anti-human cytokeratin 19 (CK19) antibody (diluted 1:300; Progen Biotechnik, Heidelberg, Germany) overnight at 4 °C. Subsequently, specimens were incubated with horseradish peroxidase (HRP) conjugated goat anti-guinea pig IgG antibody (1:200; Jackson Immuno Research, Ely, UK), washed, incubated with HRP substrate (BD Biosciences, Pharmingen, USA), and counterstained with hematoxylin. An Axio Scan Z.1 slide scanner (Carl Zeiss MicroImaging, Jena, Germany) with Zen 2013 software was used for image acquisition. The area of CK19-positive cells versus the total hematoxylin-stained area was quantified with open-source software, Fiji Image J v1.5.3.

### 2.11. Data Analysis and Statistics

Statistical analyses were performed using GraphPad Prism 8 software (GraphPad, San Diego, CA, USA). A negative binomial distribution model was applied to analyse the CRISPR screen data. For other data, unpaired *t*-test was used if they followed a normal distribution; otherwise the Mann–Whitney test was applied. Bonferroni–Hochberg adjustment was used for multiple testing.

## 3. Results

### 3.1. CRISPR-Cas9 Whole-Genome Knockout Screen

We studied PDAC cell lines S2-007 and S2-028. Both originate from the parental cell SUIT-2, whose origin was a PDAC liver metastasis [20]. They exhibit very high or low metastatic capacity, respectively, as demonstrated by in vitro studies and in vivo experiments [21,22]. While S2-007 triggers pulmonary metastasis in mice, for instance, S2-028 does not. We conducted genome-wide CRISPR-Cas9 screens in both cell lines to identify genes whose deletion was selectively cytotoxic to metastatic S2-007 cells, but not to the non-metastatic S2-028 cells. We utilized a library made of 259,900 sgRNA constructs [11]. It targets all the protein-encoding genes in the human genome with about 12 sgRNAs each, besides all the microRNA transcripts. There are also 7700 non-target control (NTC) sequences. The individual steps of the screening process are schematically shown in Figure 1A (for more details, see Appendix A). Initially, Cas9 was stably introduced into both cell lines (Appendix A). After lentiviral transduction of Cas9^+^-cells with the comprehensive knockout library and the selection of transduced cells with puromycin, cell aliquots were harvested at the beginning of the growth period (t_zero_) and after 14 cell doublings, 18 days later (t_end_). The frequency of sgRNA expression constructs inserted into the genomic DNA of the respective cell population was determined by 1000x coverage next-generation DNA sequencing (see Appendix A for all the sgRNA counts). Quality control, data analysis, and visualization were performed with MAGeCK-VISPR 0.5.6 software [15]. A negative binomial distribution model was used to test whether the sgRNA abundances differed between the metastatic and non-metastatic cells. The overall correlation between the sgRNA counts in the S2-007 (metastatic) and S2-028 (non-metastatic) samples at t_zero_ was high, at 0.97, indicating an equal distribution of the sgRNA library in both cell lines following lentiviral transduction (Figure 1B). A comparison of the t_zero_ of one cell line and the t_end_ of the other yielded a correlation of 0.85 or 0.83, respectively, documenting that changes had occurred in the two final sgRNA representations. The value of 0.92, resulting from a comparison of both t_end_ samples, finally revealed that roughly 62% of the changes occurred similarly in both cell lines, while the other 38% were specific to either S2-007 or S2-028 cells.

From the sgRNA abundance data, a beta-score was calculated that indicates to which degree a gene could be considered essential to either cell line (see Appendix A for all 22,047 transcripts). Furthermore, the beta-score variations between S2-007 and S2-028 were determined. Significant differences point at genes whose deletion could selectively affect the viability of one cell line only. At *p* < 0.05, a knockout of 590 genes significantly depleted metastatic cells; a knockout of 348 genes had the same effect on non-metastatic cells. At higher stringency (*p* < 0.01), the numbers decreased to 67 and 23, respectively (Appendix A). In particular, the 67 genes are of interest as they may have the potential to be specifically important for the survival of metastatic cells (Figure 1C). For quality control, the gene sets were subjected to a KEGG enrichment analysis. Based on the 590-gene set, enrichment in 16 KEGG pathways was found (Figure 1D). The strongest enrichment was obtained in core essential pathways [23], such as ribosome, spliceosome, and proteasome, demonstrating successful CRISPR/Cas9-mediated gene knockout and depletion of cells with deleted core essential genes. Three pathways were left when the 67 genes that had been obtained with the more stringent discrimination between metastatic and non-metastatic cells were looked at. Notably, no particular pathway was found on the basis of the genes that diminished the viability of non-metastatic compared to metastatic cells (Appendix A).

For further selection, we filtered out genes that were generally essential in both cell lines, but nevertheless assigned to the beta-score lists. One example is *RPL4*, the fifth most differential hit from the comparison of metastatic versus non-metastatic. The corresponding sgRNA counts dropped to very few or no reads for 10 of the 12 sgRNAs in both cell lines after *RPL4* knockout (Appendix A). Because of the differences for the two remaining sgRNAs and the very small overall number of counts after knockout, however, the overall difference between S2-007 and S2-028 was statistically significant, while biologically, it was probably meaningless. Because of these findings, we prioritized the identified genes further with an emphasis on genes that exhibited no or little essentiality in S2-028 while showing a substantial effect in S2-007. To this end, we calculated a drop-out value, which is the ratio between the sgRNA counts at t_end_ and t_zero_. The lower the value, the more the viability dropped because of the corresponding sgRNA. In addition, the bigger the difference between S2-007 and S2-028—with S2-028 not strongly changing—the more the gene is likely to be related to metastasis (Appendix A; Figure 2).

### 3.2. Functional Evaluation of Particular Genes

Three of the interesting candidate genes resulting from the prioritisation were *TYMS*, *MEN1*, and *MYBL2* (Appendix A). Their drop-out differences in terms of neighbouring sgRNAs were rather stable across the regions of the respective gene. This is a quality criterion, since drop-out values depend on the effectivity of sgRNA sequences, and individual sgRNAs may not function properly. *TYMS*, *MEN1*, and *MYBL2* represent different candidate types. Knocking out *TYMS* resulted in cell death in metastatic cells when early exons were targeted, while basically no differential phenotype could be observed with sgRNAs aiming further down the gene sequence (Figure 2); non-metastatic S2-028 showed only little change to survival for all the sgRNA constructs. There are other genes with similar features, such as *TRIP13*. *MEN1* knockout led to a depletion of S2-007 cells, while non-metastatic cells grew even better. The knockout of *MYBL2* finally reduced viability in both cell lines, but the effect was much more pronounced in the metastatic cells.

For further analyses, we transduced cells with sgRNAs targeting the three candidates and compared them with respect to the efficiency of the actual knockout using Western blot analyses (Appendix A). For each gene, the two best-performing sgRNAs were chosen for the functional assays. Changes in the viability of the respective cells were studied by flow cytometry, confirming the results of the genome-wide CRISPR screen. In all cases, viability was reduced more in the metastatic S2-007 cells than in the non-metastatic S2-028 cells. However, there were noteworthy differences (Figure 3A). After 14 days, the knockout of *TYMS* led to a reduction in viability by more than 50% in both S2-007 and S2-028. In contrast, the knockout of *MEN1* had little to no effect on S2-028, while the viability of S2-007 dropped by only about 25%, even after a longer incubation. Finally, *MYBL2* yielded results that fell between the two. In the non-metastatic S2-028, viability was reduced to about 75% after 21 days of incubation, while in the metastatic cell line S2-007, it was reduced by more than 50% for both sgRNAs. The analysis of cell proliferation also showed differences in cellular phenotypes. The knockout of *TYMS* reduced proliferation in both S2-007 and S2-028 (Figure 3B). The *MEN1* knockouts affected proliferation in S2-007 but not in S2-028. The *MYBL2* knockout had no clear effect on cell proliferation in either cell line.

As we aimed in particular at identifying genes that influence metastasis, colony formation capacity was studied. There was a clear distinction between *TYMS* or *MEN1* on the one side and *MYBL2* on the other. A knockout of *TYMS* or *MEN1* strongly reduced the capacity to form colonies irrespective of the cells’ metastatic capacity (Figure 4). Knocking out *MYBL2*, however, affected S2-007 drastically, while it had no significant effect on the colony formation capacity of S2-028. These results suggest that knocking out *MYBL2*, as compared to *TYMS* or *MEN1* knockouts, alters different processes and pathways.

### 3.3. Analysing MYBL2 Functions

We focused our studies on *MYBL2*, whose modulation very specifically influenced the functional aspects of metastatic tumour cells. Additionally, its expression has been shown to correlate with metastasis in several different cancer entities, such as renal cell carcinoma and breast, lung, and prostate cancer [24,25,26,27]. MYBL2 is a transcription factor of the MYB family and has a role in cell cycle progression, cell survival, and cell differentiation [28]. We conducted further functional analyses to understand its effect on processes that are important for metastasis. There were no differences in migration between the *MYBL2* knockout and the control cells for either S2-007 or S2-028 (Figure 5A). In contrast, an analysis of the cells’ invasive capabilities resulted in a clear difference. While *MYBL2* knockout in S2-007 significantly inhibited invasion, the effect was insignificant in S2-028 (Figure 5B). In summary, *MYBL2* knockout triggered changes that reduced overall cell viability, invasion capability, and colony formation capacity specifically in metastatic cells, while cell proliferation and migration were not affected.

We performed an Ingenuity Pathway Analysis (IPA) for the identification of possible interactions of MYBL2. As expected, it was shown to be an integral part of the cell cycle, interacting with regulating molecules Cbp/p300, Cdc2, cyclin A (CCNA), cyclin B1 (CCNB1), cyclin E (CCNE), E2f, FBXW7, FOXM1, MAD2L1, and UBE2C (Figure 5C). Six of the ten interacting molecules had a significant beta-score in the knockout screen comparing metastatic versus non-metastatic cells (Appendix A). Two molecules directly activated by MYBL2 are FOXM1 and CCNB1. The oncogenic forkhead box transcription factor FOXM1 in turn also activates *CCNB1*. Cyclin B1 is critical in regulating mitosis, and it is expressed predominantly during the G_2_/M phase of the cell cycle. Therefore, single and double knockouts of *MYBL2* and *FOXM1* were produced in order to study their effect. For confirmation of knockout, Western blot analyses were performed (Appendix A).

In analyses using flow cytometry, S2-007 and S2-028 cells showed a substantially different cell cycle distribution in the absence of genetic perturbations (Figure 5D). The percentage of S2-007 cells in the G_1_ phase was much higher than that of cells in G_2_/M. In comparison to S2-028, the percentage of cells in G_1_ was nearly twice as high, while the portion in the G_2_/M phase was less than half. The cell number in the S phase did not differ significantly. The single-gene or double-gene knockouts of *MYBL2* and *FOXM1* did not significantly change the percentages in S0-028. In S2-007, however, there was a strong shift from cells in the G_1_ phase to cells in G_2_/M upon knockout of *MYBL2* and/or *FOXM1*, differing substantially from unperturbed cells, while the percentage of cells in the S phase did not change significantly (Figure 5D). This is in line with the suggested regulative process involving *CCNB1* (Figure 5C), which is a regulator of the G_2_/M transition phase. Fittingly, this result is in agreement with the significant effect of a direct *CCNB1* knockout in the genome-wide CRISPR screen, upon which cell viability was, overall, reduced by a fifth more in S2-007 than in S2-028 (Appendix A).

In the cell viability assays, a *MYBL2* knockout reduced the viability of S2-028 much less than in S2-007. Additionally, a knockout of *FOXM1* and a double knockout had a substantially stronger effect in the metastatic S2-007 cells (Figure 5E). On the basis of the viability changes upon gene knockout, a genetic interaction (GI) score was calculated for *MYBL2* and *FOXM1* (Figure 5F) [29]. Genetic interaction exposes functional relationships between genes. Frequently, it is applied to reveal the effect of gene mutations [30,31], but no sequence variation was observed here. A GI score < 0 indicates synergy between two genes; GI > 0 points to a buffering genetic interaction that permits a more stable phenotype. In a 21-day period, the GI score of S2-028 was continuously negative. It was relatively stable up to day 14 and then decreased even further. Conversely, the values indicate a two-step process for S2-007. There was a steady, although weaker synergistic interaction up to day 14. Subsequently, however, the GI value took an entirely different direction, turning clearly positive, as opposed to the observed decrease in S2-028, and a solid buffering mode of genetic interaction set in.

### 3.4. In Vivo Colonisation in Mice After Intravenous or Intracardial Injection of S2-007 Cells

For analysing the effect of *MYBL2* knockout on colonization in vivo, NSG mice were inoculated with S2-007 cells in which *MYBL2* had been knocked out or that were treated with a non-target control (NTC). Six mice were studied in each group. Cells were either injected into the lateral tail vein (i.v.) or the left heart ventricle (i.card.). The animals were kept for three weeks before being sacrificed. For two and three animals in the i.v. and i.card. NTC groups, respectively, the analysis was discontinued two days earlier, since a pre-defined humane endpoint was reached. Already the overall health status showed significant differences between the NTC and knockout groups, as indicated by weight loss after three weeks (or two days earlier for some NTC animals) (Figure 6A). Tissue sections of the lungs and liver were studied by immunohistochemistry, quantifying the area of CK19-positive cells relative to the total hematoxylin-stained area (Figure 6B,C). While the observed ratios varied substantially between individual animals within each group, there were, nevertheless, clear differences between the NTC and *MYBL2* knockout in all cases apart from the livers of the i.card. mice, which showed an only insignificant difference.

## 4. Discussion

The complexity of the molecular processes responsible for pancreatic cancer poses a challenge to the thorough understanding of the disease and the establishment of new therapeutic approaches. Since metastasis is central to the disease’s quick progression and dismal prognosis, more information is required about the processes that initiate, regulate, and direct the dissemination of cells. Metastatic cells exhibit more genetic alterations than primary tumour cells [5]; the cell plasticity linked to metastasis actually implies tumour heterogeneity [32]. The two cell lines used in this study exhibit strong heterogeneity with respect to their metastatic capacity and therefore served as a good model to discover genes that are critical for cells with high metastatic potential. Since the cells were derived from one parent tumour cell line, however, overall differences in the genetic background are minimal. Nevertheless, we cannot rule out that genetic variations may have an additional effect on the difference in viability upon gene knockout.

Extending the findings to additional PDAC cell lines or patient-derived organoid models might increase the generalizability of the results. However, the SUIT-2-derived cell models allowed us to rigorously investigate factors related to metastatic potential within a controlled experimental framework. Therefore, we deliberately decided against an analysis of other cell lines. Instead, we used initial animal experiments. Besides avoiding the variation and frequently undefined status of metastasis capacity in other tumour cell lines, the animal analysis was more indicative of the relevance for metastasis in a complex biological system.

In the CRISPR-Cas9 screen, we looked at all human genes in a redundant manner. Its results represent basic information, linking functional variations to molecular processes, and provides a starting point towards a better understanding of the molecular background of phenotypical changes [11,33,34]. If a significance threshold of *p* < 0.05 was applied, the screen identified 590 genes that affect the viability of metastasizing cells more than that of non-metastasizing cells. This relatively large number implies that a mixture of factors or regulative pathways is likely to be responsible for relevant changes. In consequence, this implies the possibility of influencing metastasizing cells in different ways that may complement each other. Conversely, no single target is likely to exhibit an effect that is strong enough to stop metastasis altogether.

Our CRISPR analysis was limited by the inherent drawbacks of two-dimensional cell culturing. While providing valuable insights into metastasis-related genes in PDAC, the cell culture model does not fully capture the complex interactions between tumour cells and tumour microenvironment, which plays a crucial role in cancer biology [35]. In vivo screens could take care of this aspect [36]. Because of the very large number of sgRNA constructs, however, performing a solid and informatively comprehensive CRISPR screen was quite demanding, even in vitro. The number of mice that would have been required for an equivalent in vivo analysis was such that performing a screen of this kind without prior knowledge on possibly relevant genes would have been technically and ethically challenging. We therefore opted for selecting potential genes of interest on the basis of in vitro screening. An in vivo assay of the 590 or 348 genes significantly depleted in metastatic or non-metastatic cells, respectively, would be substantially easier to implement.

The utilisation of, on average, 12 sgRNA constructs indicated differences in the drop-out values along the length of the coding sequence of quite a few genes. Some of the changes are probably intrinsic to the methodology and down to the purely technical aspect that individual constructs exhibit varying knockout performances. However, genes such as *TYMS* or *TRIP13* highlight the fact that differences are likely to also be a consequence of the position at which the actual knockout took place. There were several genes exhibiting variations along the gene sequence that were not just measured with one knockout construct, but were reproduced by neighbouring sgRNAs. One reason could be splice variants, for example, which could affect knockdown efficiency and, thus, the drop-out value. In turn, basic information could be derived about gene structure and function by analysing in detail the variations of individual drop-out values.

We concentrated our functional analyses on a few of the gene candidates. One of them was *TYMS*, which is active in the pathway for the de novo production of deoxythymidine monophosphate. TYMS is the main target of 5-fluoruracil (5-FU), one of the drugs used to fight PDAC. Elevated expression of *TYMS* has been linked to resistance to 5-FU [37]. Furthermore, high *TYMS* levels are associated with poor overall and shortened recurrence-free survival [38], indicating their importance to tumour subsistence and metastasis. With respect to *MEN1*, mutations in or the deletion of the gene have been reported to play a role in pituitary tumorigenesis, with the gene apparently acting as a tumour suppressor [39]. *MEN1* silencing is involved in the metastasis of prostate cancer [40], supporting the findings of our study. Concerning PDAC, *MEN1* was recently found to be an important mediator of homeostasis in vivo, and the loss of *MEN1* impaired regeneration [41]. Our analysis of *MYBL2* confirmed its known involvement in the regulation of checkpoints of the cell cycle. *MYBL2* knockout has been reported to reduce cell proliferation and decrease the percentage of cells in the G_2_/M phase [28]. In contrast to this, we did not find any major effect of *MYBL2* knockout on either proliferation or cell cycle distribution in S2-028 cells. S2-007 proliferation was also not affected. The parent cell line, SUIT-2, originated from metastasis, while many other cells in which *MYBL2* knockout was studied were derived from primary tumours or non-cancerous tissues. This may explain the lack of proliferation changes in the cells used in this study, despite the fact that it was observed in others. Furthermore, it could be possible to detect a difference in cell proliferation in a three-dimensional system (i.e., spheroids, organoids) that is missed in a two-dimensional system.

The knockout of *MYBL2* in S2-007 cells triggered a strong shift towards G_2_/M. The *MYBL2* and *FOXM1* double knockout did not have additive effects on G2/M cell cycle transition compared to the single deletions. However, the double knockout did exhibit a more pronounced impact on PDAC cell viability, suggesting that the functional consequences of *MYBL2* and *FOXM1* deletion in PDAC cells are mediated by mechanisms beyond the direct regulation of the cell cycle. One possible explanation could be that MYBL2 and FOXM1 might influence cancer cell survival through mechanisms other than cell cycle regulation, such as apoptosis or metabolic adaptation, which are critical for sustaining the growth and survival of metastatic cancer cells. These processes might be differentially affected by the combined loss of both transcription factors, thus accounting for the observed reduction in viability. Further investigation into these alternative pathways could help elucidate the broader functional roles of the interaction between MYBL2 and FOXM1 in PDAC metastasis.

A look at processes that take care of DNA damage may provide an explanation for the increase in the number of S2-007 cells in G_2_/M. DNA damage is dealt with at both the G_1_ and the G_2_ checkpoints of the cell cycle [42]. The former is regulated by p53 activity. Because of the frequently mutated *TP53* status in PDAC, this regulative process is circumvented, however. Upon metastasis, overexpression of *MYBL2* apparently overrides, additionally, the G_2_ checkpoint in a p53-independent manner [43,44], while this is not observed in non-metastatic cells. G_2_/M transition is regulated by the interactions of protein multi-vulva class B (MuvB) [45]. During G_0_ and G_1_, MuvB interacts with other proteins forming the DREAM complex, which represses transcription. The DREAM-specific proteins dissociate from MuvB at late G_1_, which is then free to associate with MYBL2 during the S phase. The MYBL2-MuvB complex recruits FOXM1 in G_2_ [28,46,47], triggering the expression of genes such as *CCNB1*, which is required for G_2_-to-M transition [48]. Variations in the concentration of the complex components influence the process and lead to abnormalities, such as an arrest in G_2_/M upon *MYBL2* knockout.

The MuvB complex formation process may also explain the observed buffering interaction effect of *MYBL2* and *FOXM1* (Figure 7). MYBL2 is required for the recruitment of FOXM1 into the complex and its subsequent target gene binding, but it is no longer needed once the complex has been formed [28,49,50]. MuvB-FOXM1 exhibits activity without MYBL2, thus buffering the activity of the MYBL2-MuvB complex. This is supported by the viability assays with individual knockout constructs. The effect of the *FOXM1* knockout on cell viability was less severe as compared to that of the *MYBL2* knockout. However, it led to the same functional changes. The buffering also explains why *FOXM1* was not found as a metastasis-specific gene in the CRISPR-screen, while *CCNB1*—a downstream gene of both *MYBL2* and *FOXM1*—was actually identified as such. MYBL2 is required to trigger the activity of FOXM1 by facilitating the formation of a complex with MuvB. However, FOXM1 is not needed to form the MYBL2-MuvB complex. Therefore, MYBL2 is essential for activating *CCNB1*, while FOXM1 is not. Fittingly, *CCNB1* has been shown to significantly affect the EMT characteristics of pituitary adenomas [51]. Furthermore, *MYBL2* knockdown suppresses cell invasion, anchorage-independent growth. and tumour formation by acting as a regulator of SNAIL, a protein that promotes EMT [52]. This process is also regulated via FOXM1 [53] in the same way as other cancer-related genes, including *TYMS* [54], another major protagonist in our screen that was found to be vital to metastatic cells.

## 5. Conclusions

The comprehensive CRISPR knockout screen revealed a substantial number of genes that are associated with the metastatic behaviour of pancreatic cancer cells. Taking advantage of this knowledge could result in a better understanding of processes that affect metastatic cells but do not influence non-metastatic cells significantly. By looking at three of the identified genes, different processes and regulative pathways were suggested, indicating the wide range of molecular functions involved in metastasis. For the well-known transcription factor MYBL2, another mode of regulating its activity was suggested that makes metastasising cells different from non-metastasising ones. A change from a synergistic to a buffering interaction mode in *MYBL2* and *FOXM1* could be a factor that is critical to PDAC metastasis. This finding may offer new opportunities for therapeutic intervention. More experimental analyses and confirmation are required for a mechanistic understanding of the apparently many processes involved in metastasis. The identification in our screen of a substantial number of genes involved in these processes will facilitate these investigations.

## Figures and Tables

**Figure 1 cancers-16-03684-f001:**
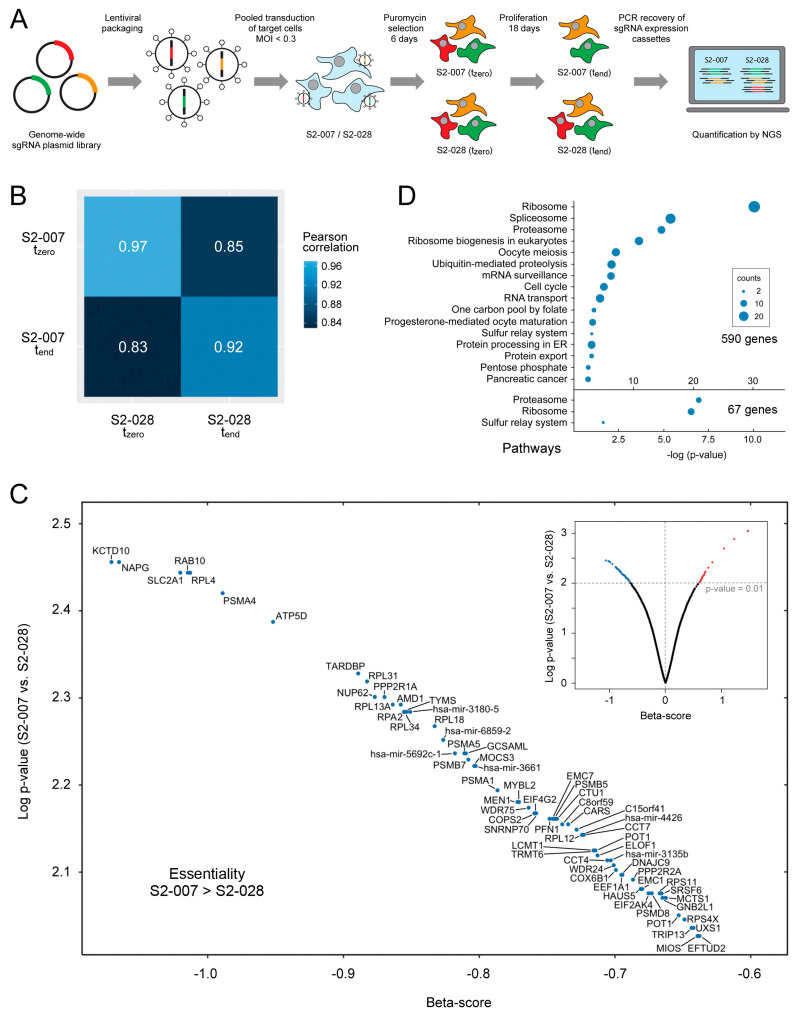
CRISPR-Cas9 knockout screen results. (**A**) Scheme of the CRISPR-Cas9 screening process. (**B**) Heatmap of the pairwise Pearson correlations of samples’ log sgRNA counts (*p* < 2.2 × 10^−16^). ((**C**) bottom panel) The distribution of the beta-scores for all 259,900 sgRNA constructs is shown in the small inserted display. Genes whose knockout led to a reduction in the abundance of metastatic cells as compared to non-metastatic cells (*p* < 0.01) are shown enlarged (blue dots). Red dots represent genes that led to a significant reduction of non-metastatic cells. Black dots stand for cells which showed no significant change in abundance. ((**D**) centre-right panel). Pathways resulting from a KEGG enrichment analysis of the genes whose knockout most reduced the viability of metastatic cells. Results are shown from analysing 590 genes (*p* < 0.05; top) and 67 genes (*p* < 0.01; bottom).

**Figure 2 cancers-16-03684-f002:**
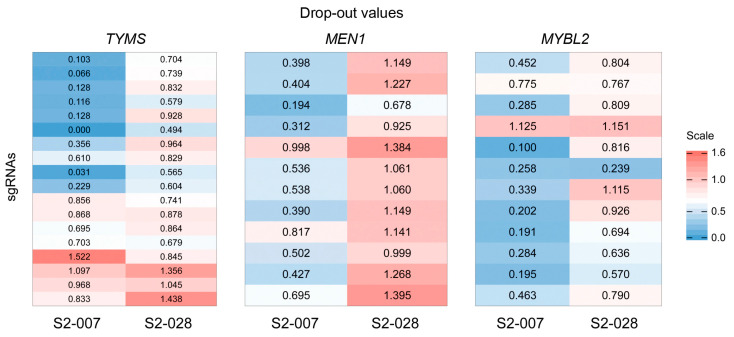
Knockout screening results for genes *TYMS*, *MEN1*, and *MYBL2*. Drop-out values—ratios of the sgRNA counts at t_end_ and t_zero_—were calculated for all sgRNAs of the candidate genes. They are shown in the order of their genomic location and color-coded. Blue indicates a clear decrease, while white represents small or no changes; red stands for a strong increase.

**Figure 3 cancers-16-03684-f003:**
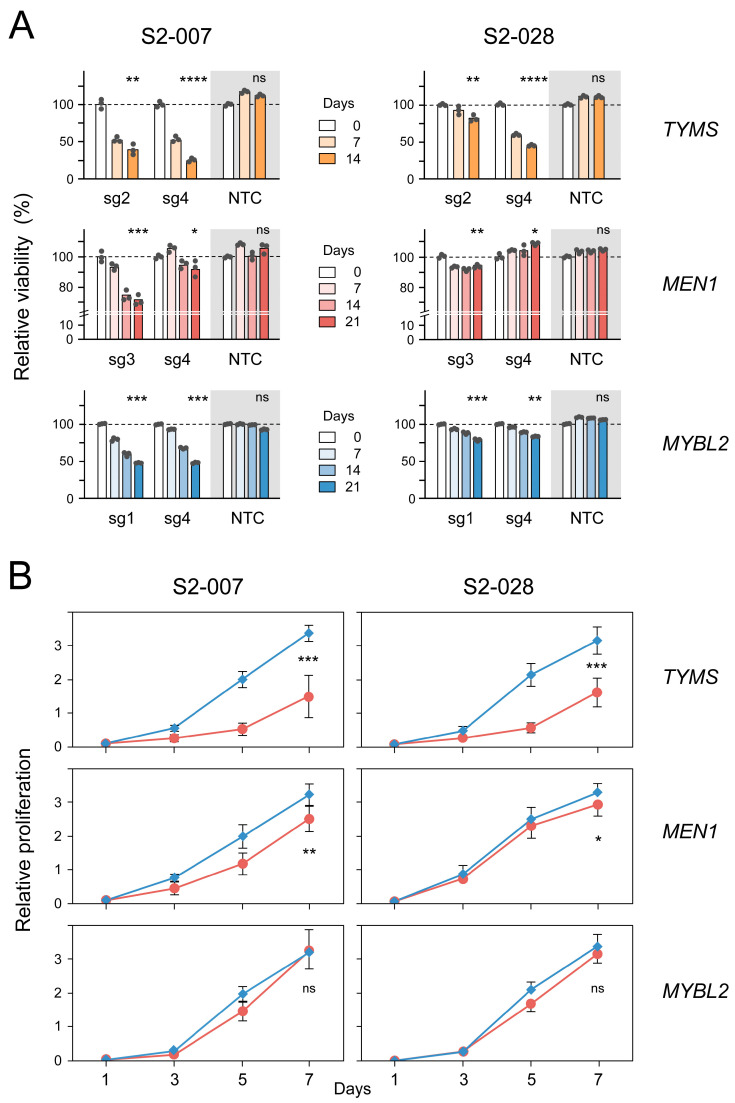
In vitro validation of candidate genes *TYMS*, *MEN1*, and *MYBL2*. (**A**) Cell viability in S2-007 and S2-028 cells was determined by flow cytometry. For each gene, the two best-performing sgRNAs (sg) were used. The shaded area highlights the NTC control. (**B**) Changes in proliferation are shown as mean ± standard deviation. Each red circle indicates the result of 40 growth experiments (eight each of five sgRNAs). Blue squares represent 24 control measurements (eight growth analyses each of 2x NTC and 1x vector). Data for the individual knockouts and controls can be seen in Appendix A. **** = *p* < 0.0001; *** = *p* < 0.001; ** = *p* < 0.01; * = *p* < 0.05; ns = not significant.

**Figure 4 cancers-16-03684-f004:**
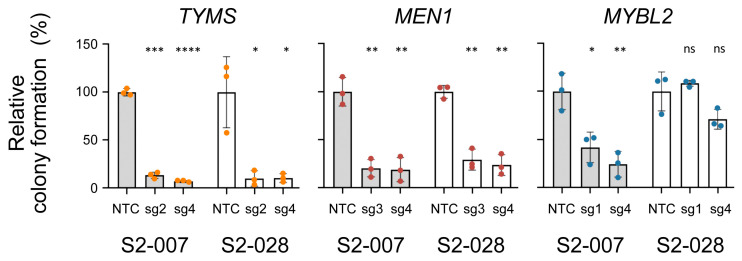
Measurement of colony formation capacity. Data are presented as mean ± standard deviation; NTC stands for non-targeting control. Grey columns: S2-007; white columns: S2-028. **** = *p* < 0.0001; *** = *p* < 0.001; ** = *p* < 0.01; * = *p* < 0.05; ns = not significant.

**Figure 5 cancers-16-03684-f005:**
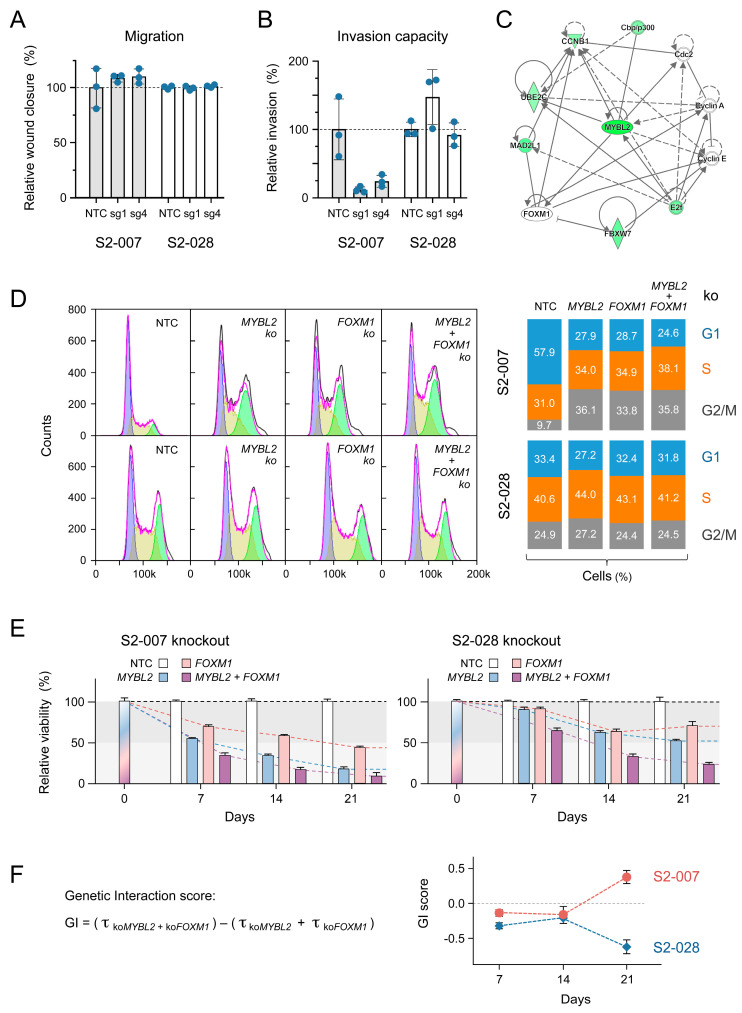
Characterisation of functional effects of *MYBL2* knockout. (**A**) Cell migration was analysed with the two best-performing sgRNAs (sg); NTC stands for non-target control. (**B**) Additionally, the invasion capacity of the cells was studied. (**C**) MYBL2 interaction map resulting from an IPA analysis of the knockout data. Solid lines stand for direct, dotted lines for indirect interactions; green colouring indicates molecules that exhibited in the genome-wide screen a reduced presence upon knockout. (**D**) Changes in cell cycle phase distribution in S2-007 and S2-028 cells upon single or double knockout, as indicated; NTC, non-target control. Representative flow cytometry histograms (left) and the overall percentages (right) are shown. (**E**) Viability measurement by flow cytometry of cells in which *MYBL2* and *FOXM1* were knocked out individually or together. Data are presented as mean ± standard deviation (n = 3). (**F**) Based on the above data, GI scores were calculated for days 7, 14, and 21. Each value represents the mean of three independent measurements.

**Figure 6 cancers-16-03684-f006:**
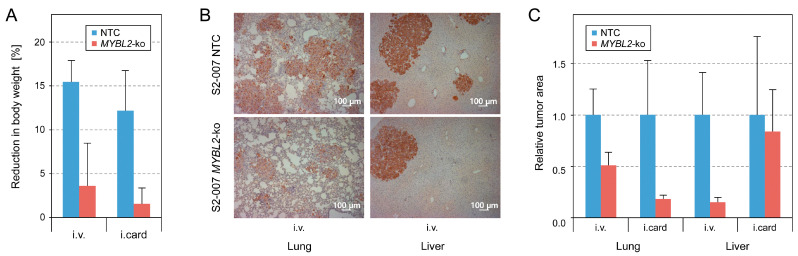
In vivo analysis of colonisation. (**A**) S2-007 cells with *MYBL2* knockout or a non-target control (NTC) were injected into the lateral tail vein (i.v.) or the left heart ventricle (i.card.) of NSG mice. Loss of body weight was recorded at the time of experiment discontinuation; there were six mice in each group. (**B**) Lung and liver sections were studied by CK19 immunohistochemistry; typical results are shown. (**C**) A ratio was calculated of the area of CK19-positive cells relative to the total hematoxylin-stained area. The median and the median absolute deviation are shown for each group of mice.

**Figure 7 cancers-16-03684-f007:**
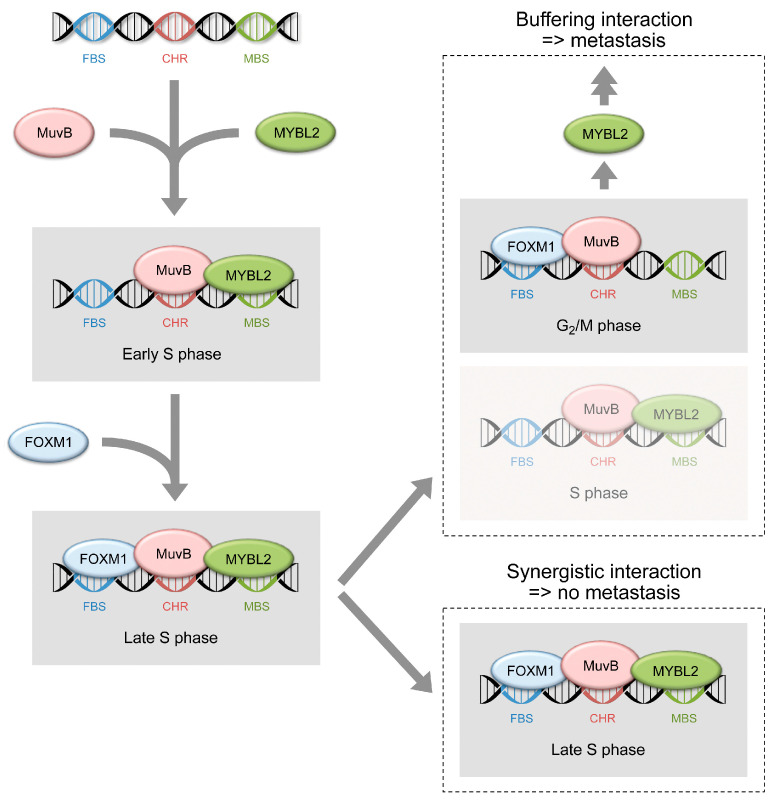
Graphical representation of a proposed MYBL2-and-FOXM1 interaction process. The observed differences between non-metastatic (synergistic interaction) and metastatic cells (buffering interaction) are shown. FBS: FOXM1 binding site; CHR: cell cycle genes homology region; MBS: MYBL2 binding site.

## Data Availability

The original contributions presented in this study are included in the article and Appendix A. Further inquiries can be directed to the corresponding author.

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
