# Peer review of "Genome-Wide CRISPR Screen Identifies Genes Involved in Metastasis of Pancreatic Ductal Adenocarcinoma"

_cancers, 2024, doi:10.3390/cancers16213684_

Round 1

Reviewer 1 Report

Comments and Suggestions for Authors

This article identifies genes associated with metastasis in pancreatic ductal adenocarcinoma (PDAC) through genome-wide CRISPR screening. The study revealed that the knockout of 590 genes significantly impacted the survival of PDAC metastatic cells, while having no effect on non-metastatic cells. Further functional investigations demonstrated alterations in the genetic interaction between MYBL2, a gene associated with metastasis, and FOXM1, which is crucial for PDAC metastasis. This discovery presents new prospects for therapeutic intervention and serves as a foundation for further exploration into multiple molecular functions involved in metastasis. The manuscript is overall of high quality and can be accepted with minor revisions. The following suggestions for modification are provided.

1. The introduction section would benefit from additional descriptive content, typically consisting of 3-4 paragraphs.

2. The following references are recommended due to their relevance to the author's topic and their potential to enhance the manuscript's quality.

[1] Li H, Wei WY, Xu HX. Drug discovery is an eternal challenge for the biomedical sciences. Acta Materia Medica. 2022, 1(1): 1-3. DOI: 10.15212/AMM-2022-1001

The literature listed below can be referenced in an introductory section on the tumor microenvironment, following citation 31.

[2] A. Gu, J. Li, S. Qiu, S. Hao, Z.-Y. Yue, S. Zhai, M.-Y. Li, Y. Liu, Pancreatic cancer environment: From patient-derived models to single-cell omics. Mol. Omics 2024, 20, 220233. DOI: 10.1039/D3MO00250K

3. The authors should provide a rationale for their selection of S2-007 and S2-028 cells.

4. The author should utilize P-values to indicate the disparities in Figure 3 and Figure 4.

5. There is a necessity to enhance the depiction of genetic impact on phenotypic traits, encompassing cell proliferation, growth, migration, invasion and so forth.

6. The authors should explicitly acknowledge a limitation of the study, specifically the absence of organoid models to investigate the impact of the tumor microenvironment.

Author Response

All responses to the reviewer's comments are given in the attached pdf-file.

Reviewer 2 Report

Comments and Suggestions for Authors

The authors conducted a genome-wide CRISPR cas9 knockout screen to investigate genes that differentially affect metastatic vs nonmetastatic pancreatic cancer cells. Upon comparison of essential genes for metastatic vs nonmetastatic pancreatic cancer cells, the authors identified 67 potential hits, from which 3 were further validated, namely TYMS, MEN1 and MYBL2. Single clones of these hits were then established, viability and proliferative capabilities were subsequently evaluated. The authors found that MYBL2 selectively affected metastatic S2-007 cells viability but not proliferation or migration. IPA analysis indicated that MYBL2 regulates cell cycle and activates FOXM1 and CCNB1. The authors argued that FOXM1 may be buffered by genetic interaction, which would explain why it didn't show up as a hit from their CRISPR KO screen. Lastly, the authors validated their findings in vivo, and found that MYBL2 KO reduced tumor volume in NSG mice. 

The manuscript could improve by addressing the following: 

1. Did the authors validate their findings in other PDAC cell lines not originated from SUIT-2? Both genetic interaction and signaling may be context/cell line dependent, therefore if the findings also hold true for other cell line backgrounds, the significance and novelty of the finding is significantly increased.

2. There is a lack of quality control figures for the CRISPR screen, such as sequence depth, which can be demonstrated by plotting log(reads) per sample, or reference core essential vs nonessential genes. 

3. In Fig 2, vertically labeling the targeted regions for each gene (i.e. exon, introns etc), may make this figure more informative. Additionally, is there any specific reason there are different numbers of sgRNA for each target?

4. Line 307-317: The authors mentioned that MYBL2 regulates cell cycle, what does the authors consider to be the potential reason that MYBL2 KO showed no effect on proliferation in their study?

5. Did the authors examine the regulation of FOXM1 by MYBL2 in their models? Are there mRNA/protein level changes of FOXM1 following KO or OE of MYBL2?

6. Fig 5D: no legend for the left panel

Author Response

(The authors gave the same response as above.)

Reviewer 3 Report

Comments and Suggestions for Authors

I would like to complement the author (s) for conducting the study implementing CRISPR-Cas9 gRNA screen and identifying gene set contributing overall growth of metastatic cells. 

In the data, the combined deletion of MYBL2 and FOXM1, there is not much difference in promoting G2/M transition between single deletion. Can Author (s) explain about that? And how much of that difference could be accounted to author (s) hypothesis?

Maybe, I did not notice, can author explain what experimental sets were derived from HEK293T cells, and any specific reason to grow them in IMDM instead of routinely use DMEM.

- In the entire manuscript, author (s) tends to validate the effects of MYBL2 on proliferation or viability. Can author (s) explain, what new information the current manuscript will add about MYBL2, while its metastasis and supporting roles have been well reported in various cancer studies (Mol Cancer Res (2020) 18 (2): 311–323, DOI: 10.1186/s12885-020-07135-2, DOI: 10.1007/s10735-020-09920-6).

Author Response

(The authors gave the same response as above.)

Reviewer 4 Report

Comments and Suggestions for Authors

Dr. Hoheisel and his team published an article using a genome-wide CRISPR screen to identify pancreatic ductal adenocarcinoma (PDAC) metastasis genes. This is a well-written article with significant translational impact. However, a few issues must be addressed before acceptance can be considered. These are as follows:

1. It is well established that oncogenic KRAS is a major driver gene in PDAC (PMID: 33870211). Recently, genome-wide screens have also demonstrated that Yap/Hippo signaling plays a role in generating drug resistance in KRAS-driven cancers (PMID: 37729426). Furthermore, MYBL2 has been implicated in mediating drug resistance through Hippo signaling (PMID: 33897882), while FoxM1 has been shown to mediate cross-talk between Kras-regulated signaling pathways (PMID: 22826436). Given this, it is reasonable to speculate that Hippo signaling may play a significant role in PDAC metastasis through inter-signaling regulation with MYBL2/FOXM1 pathways. The authors should include a brief discussion of this potential role by citing the relevant studies mentioned, as this could be a key direction for future research on PDAC metastasis.

2. For Figure 4, please perform a one-way ANOVA followed by Tukey's multiple comparisons test, and include this information in the figure legend.

3. The authors should include a graphical representation of the CRISPR screen, illustrating how the screen was conducted. A schematic of the CRISPR screening strategy should be added to Figure 1. 

Comments on the Quality of English Language

The overall English is good, but the authors should refine the introduction and discussion sections before it is ready for acceptance. In the discussion section, the statements should be presented in direct voice.

Author Response

(The authors gave the same response as above.)

Round 2

Reviewer 3 Report

Comments and Suggestions for Authors

Please ensure the changes has been implemented. In the submitted manuscript, S2 cells are still stated to be grown in DMEM, and HEK293T cells in IMDM media. Please correct that.

Reviewer 4 Report

Comments and Suggestions for Authors

All concerns addressed. 
